# Hyperbaric Oxygen in Post-Stroke Patients: A Feasibility Study

Jörg Schmutz [1,*], Stefan Engelter [2,3,*], Nils Peters [2,3,*], Patrick Schmucki [2,3] and Marco Gelsomino [1]

1.  Druckkammerzentrum Basel, Kleinhüningerstrasse 177, 4057 Basel, Switzerland; marco.gelsomino@hin.ch
2.  Neurology and Neurorehabilitation, University Department of Geriatric Medicine FELIX PLATTER, University of Basel, 4003 Basel, Switzerland; patrick.schmucki@stud.unibas.ch
3.  Stroke Center, Department of Neurology, University Hospital Basel, University of Basel, 4003 Basel, Switzerland
*   Correspondence: joerg.schmutz@hin.ch (J.S.); stefan.engelter@usb.ch or stefan.engelter@felixplatter.ch (S.E.); nils.peters@usb.ch or nils.peters@felixplatter.ch (N.P.)

**Abstract:** Background: Hyperbaric oxygen therapy (HBOT) has been shown to improve the outcomes of certain stroke patients. Our objective was to assess the feasibility of employing HBOT in daily practice in unselected stroke patients with mild-to-moderate residual post-stroke symptoms, considering their ability to commute our center. Methods: This was an exploratory, interventional, prospective monocentric study on post-stroke patients who have completed their in-hospital stroke rehabilitation. We aimed to include 10 participants who were able to complete 40 daily HBOT sessions (2.0 ATA). Effectiveness was assessed using the National Institutes of Health Stroke Score (NIHSS) pre- and post-HBOT. Results: We recruited 13 patients (12 males) with a mean age of 61 years. Three patients dropped out (two never started HBOT and one withdrew after five sessions because of traveling distance). Post-stroke time was 4–251 months. Among the 10 patients completing the HBOT program, 8 improved their NIHSS by a mean of 1,3 (1–4), while 2 patients' NIHSS remained unchanged. There were no serious adverse events and no side effects. Conclusions: HBOT was shown to be feasible for mobile post-stroke patients who have completed standard rehabilitation. In the absence of major safety concerns, HBOT seems to be an interesting option post-stroke, with the potential to further improve residual stroke severity.

**Keywords:** post-stroke; hyperbaric oxygen therapy; feasibility; outpatient

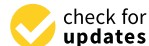



## 1. Introduction

Hyperbaric oxygen therapy (HBOT) is a well-known treatment for many conditions, as listed by the European Committee for Hyperbaric Medicine [1]. In recent times, much has been discovered regarding its mode of action, with high oxygen flow improving oxygen delivery to hypoxic tissues, suggesting that its action is mainly dependent on the free oxygen radicals that are produced. These free oxygen radicals have a significant positive impact on cellular processes, transcription factors, mitochondrial function, oxidative stress and inflammation [2]. Long courses of HBOT have been shown to stimulate neurogenesis and reduce inflammation after ischemic stroke in an animal model [3]. HBOT has therefore also been proposed for the treatment of chronic stroke in humans. A randomized controlled trial has shown both brain activity, as measured with SPECT, and corresponding motor functions improved in patients receiving HBOT as compared with the control group [4]. In a more recent case study, HBOT induced improved inter-hemispheric connectivity in a patient's brain, as shown with fMRI [5].

In Switzerland, there are only two hyperbaric facilities [6] functioning in accordance with the European Code of Good Practice (ECGP) for HBOT [7]. Therefore, our purpose was to assess the feasibility of implementing HBOT in daily practice for stroke patients who were left with disabling stroke sequalae, despite an intensive and comprehensive rehabilitation program, and who were interested in innovative means of further improving their neurological

impairments. We wanted also to know if these patients were sufficiently motivated and able to travel daily from another city, with traveling distances of up to 110 km.

## 2. Materials and Methods

After approval from the local ethics committee (EKNZ, Nr. 2017-02131), patients with stroke sequelae who had completed a standardized in-hospital stroke rehabilitation program were asked if they were interested in participating in HBOT as an additional therapeutic approach. Our primary objectives were to assess the feasibility of providing HBOT to chronic stroke patients, review patient compliance with the interventional procedures and evaluate safety issues. Secondary objectives included examination of improvements in (i) severity of post-stroke symptoms and (ii) mobility as measured 3 months after the HBOT, as compared with baseline.

Patients were recruited through our family practice and by our neurologist (SE, NP, PS). Table 1 shows the Inclusion/Exclusion criteria. Patients were allowed to have concurrent physiotherapy if this was considered useful by by the currently treating physicians. These physiotherapy programs were separate from the HBOT program and were not considered to alter the impact of HBOT. All patients who met the eligibility criteria were invited to participate in this study. After a successful ear pressure test at 1.5 atmospheres absolute (ATA), patients were asked to sign an informed consent form for the HBOT study. Tympanic tubes were not considered if a patient failed the pressure test, and these patients were not considered eligible for this study. Some patients had to travel from another city to be treated in our outpatient center. Recruitment (2018–2021) was stopped after 10 patients successfully completed HBOT.

**Table 1.** Inclusion/Exclusion criteria.

| Inclusion | Exclusion |
| --- | --- |
| Ischemic stroke with residual defects | Inability to understand the informed consent form |
| Conclusive brain CT or brain MRI | Inability to put on a face mask |
| Uneventful pressure test at 1.5 ATA | Claustrophobia |
| | Uncontrolled epilepsy |
| | Dementia or psychological disorder |
| | Previous pneumothorax |
| | Positive pregnancy test |
| | No contraception measures taken for the duration of HBOT |
| | Unsuccessful pressure test at 1.5 ATA |
| | Missed more than 10 consecutive HBOT sessions |
| | Completed fewer than 30 HBOT sessions |

All patients were treated in our multiplace hyperbaric chamber at a pressure of 2.0 ATA, according to ECGP. Each treatment lasted 2 h with 90 min of oxygen and a 5 min air break after 45 min, as a prevention against neurological toxicity due to high-pressure oxygen inhalation. Oxygen was provided with a fitted face mask.

We measured potential visual side effects in order to assess whether these would present differently in our elderly neurological patients compared to the literature. For this, we used the Snellen chart before the patient commenced HBOT, immediately after 40 HBOT sessions and 3 months after the last HBOT session.

Possible ear, nose and throat (ENT) side effects were assessed with a one-minute interview and a digital otoscopy before and after every HBOT session to determine whether otoscopy could be used to detect a barotrauma before the patient became symptomatic, thereby necessitating a pause in HBOT. The assessment was carried out by the treating

hyperbaric physician, who was present during HBOT in accordance with ECGP. Patients with persistent ENT side effects were offered tympanic tubes.

Stroke severity and tracking were quantified using the National Institutes of Health Stroke Scale (NIHSS) score at baseline and 3 months after HBOT, and both assessments were performed by an experienced stroke neurologist (SE) in an outpatient setting. A further neurological assessment was carried out by the hyperbaric physician using the Rivermead mobility index (RMI) before the start of HBOT, then weekly and again 3 months after the completion of HBOT. Patient compliance was assessed daily, and a hyperbaric technician completed the attendance list.

Stroke severity as measured by the NIHS scoring system was categorized as follows:

- 01–04 = minor stroke;
- 5–15 = moderate stroke;
- 15–20 = moderate/severe stroke;
- 21–42 = severe stroke.

## 3. Results

The participating patients, including 1 woman and 12 men, had a mean age of 61 years (age range), and the mean time from stroke to the start of HBOT was 43 months. Ten patients completed the HBOT program, with a mean age of 60 years and a mean time from stroke to the start of HBOT of 52 months (31 months if we exclude patient 2; see Table 2).

**Table 2.** Details of patient compliance.

| Patient | Total Missing Days | Pattern of Missed Sessions (Days) | Reason for Missed Sessions |
|---------|--------------------|-----------------------------------|----------------------------|
| 1 | 2 | $2 \times 1$ | Hyperbaric center was closed |
| 2 | 7 | $2 \times 1, 1 \times 5$ | Hyperbaric center was closed, holidays |
| 3 | 6 | $4 \times 1, 21 \times 2$ | Personal reasons |
| 4 | 11 | $3 \times 1$, stopped after 32 HBOT sessions | Personal reasons, stopped because HBOT had no effect |
| 6 | 0 | No missing sessions | |
| 9 | 10 | $1 \times 1, 1 \times 4, 1 \times 5$ | Personal reasons, holidays |
| 10 | 15 | $2 \times 1, 1 \times 2, 2 \times 3, 1 \times 5$ | Hyperbaric center was closed, Interruptions because of barotrauma and long traveling distance |
| 11 | 4 | $4 \times 1$ | Personal reasons |
| 12 | 4 | $4 \times 1$ | Personal reasons |
| 13 | 25 | $11 \times 1, 2 \times 2, 2 \times 5$ | Had acute COVID-19 infection, long traveling distance, other personal reasons |

### 3.1. Compliance

A total of 13 patients were recruited, and Table 2 illustrates patient compliance with scheduled HBOT sessions.

Patient 5 started HBOT as scheduled, but after five sessions, the patient indicated that commuting to our center had become too time-consuming and they decided to withdraw from this study.

Patients 7 and 8 signed the informed consent; however, Patient 7 did not present for the first neurological examination, and while Patient 8 was examined by the neurologist, the patient did not present for HBOT.

In total, 10 out of 11 patients who started HBOT completed 40 HBOT sessions, as prescribed.

All patients except patient 4 fulfilled 40 sessions of HBOT.

*3.2. Neurology*

Tables 3 and 4 show the results of the neurological assessments for all patients. Eight of the ten patients who completed this study showed some improvement in NIHSS with HBOT, while in two patients, there was no effect. Improvement was also seen in one patient more than 20 years after his or her stroke. There was no improvement in self-reported RMI, and there were no signs of acute oxygen brain toxicity.

**Table 3.** Effect of 40 sessions of HBOT at 2.0 ATA on rehabilitated post-stroke patients.

| Patient # and (Age) | Time since Stroke (Months) | NIHSS before HBOT | NIHSS 3 Months after HBOT | RMI before HBOT | RMI after 10 HBOT Sessions | RMI after 20 HBOT Sessions | RMI after 30 HBOT Sessions | RMI after 40 HBOT Sessions | RMI 3 Months after HBOT Sessions |
|---|---|---|---|---|---|---|---|---|---|
| 1 (57) | 65 | 4 | 3 | 14 | 14 | 14 | 14 | 14 | 14 |
| 2 (86) | 251 | 3 | 2 | 13 | 14 | 14 | 14 | 14 | 14 |
| 3 (63) | 60 | 5 | 2 | 15 | 14 | 15 | 15 | 15 | 15 |
| 4 (69) | 14 | 8 | 8 | 9 | 9 | 6 | 8 | - | 8 |
| 5 (58) | 4 | 3 | withdrawn | | | | | | |
| 6 (75) | 12 | 4 | 4 | 12 | 13 | 13 | 13 | 15 | 15 |
| 7 (60) | 19 | withdrawn | | | | | | | |
| 8 (69) | 7 | 6 | withdrawn | | | | | | |
| 9 (65) | 18 | 4 | 3 | 19 | 14 | 15 | 15 | 15 | 15 |
| 10 (35) | 47 | 6 | 2 | 15 | 15 | 15 | 15 | 15 | 15 |
| 11 (66) | 15 | 3 | 2 | 13 | 13 | 13 | 14 | 14 | 14 |
| 12 (57) | 21 | 3 | 2 | 15 | 15 | 15 | 15 | 15 | 15 |
| 13 (34) | 17 | 4 | 3 | 15 | 15 | 15 | 15 | 15 | 15 |

**Table 4.** Details of NIHSS changes.

| Patient | Changes in NIHSS 3 Months after 40 HBOT Sessions | Patient Self-Reports |
|---|---|---|
| 1 | NIHSS 4 ↗ 3<br>facial palsy 1 ↔ 1<br>left arm motricity 2 ↗ 1<br>left leg motricity 1 ↔ 1 | Less fatigue; improved supination. |
| 2 | NIHSS 3 ↗ 2<br>ataxia extremities 2 ↔ 2<br>dysarthria 1 ↗ 0 | Psychological state very good; left arm better; walking unchanged. |
| 3 | NIHSS 5 ↗ 2<br>facial palsy 1 ↔ 1<br>right leg motricity 1 ↔ 1<br>right leg ataxia 2 ↗ 0<br>sensibility 1 ↗ 0 | Did not know if it was helpful; sensibility clearly better. |
| 4 | NIHSS 8 ↔ 8<br>orientation 1 ↔ 1<br>visual field 2 ↔ 2<br>motricity 1 ↔ 1<br>ataxia 1 ↗ 0<br>sensibility 0 ↘ 1<br>language 1 ↔ 1<br>dysarthria 1 ↔ 1<br>neglect 1 ↔ 1 | |
| 6 | NIHSS 4 ↔ 4<br>left leg motricity 1 ↔ 1<br>left leg ataxia 2 ↔ 2<br>left sensibility 1 ↔ 1 | Patient's impression of improved walking. |

**Table 4.** *Cont.*

| Patient | Changes in NIHSS 3 Months after 40 HBOT Sessions | Patient Self-Reports |
|---|---|---|
| 9 | NIHSS 4 ↗ 3<br>facial paresis 1 ↗ 0<br>right leg motricity 1 ↔ 1<br>right ataxia 1 ↔ 1<br>dysarthria 1 ↗ 0 | |
| 10 | NIHSS 6 ↗ 2<br>right leg and arm motricity 1 ↗ 0 | Patient's impression of improving; did not know if it was due to HBOT. |
| 11 | NIHSS 3 ↗ 2<br>right leg motricity 1 ↔ 1<br>right leg ataxia 2 ↗ 1 | Found treatment very good and felt that right leg had improved. |
| 12 | NIHSS 3 ↗ 2<br>right leg motricity 1 ↗ 0<br>right leg ataxia 1 ↗ 0 | The patient saw no effect of HBOT; function of the hand had improved slightly but did not correlate with the NIHSS. Handwriting was slightly improved. |
| 13 | NIHSS 4 ↗ 3<br>right arm motricity 1 ↗ 0 | No effect on tiredness; right arm felt heavy but did not drop when held in horizontal position. |

↔ = unchanged, ↗ = improved, ↘ = deteriorated/worse.

### 3.3. Visual Acuity

Table 5 shows the results from the visual side effect assessments. We could not detect a shift towards HBOT-induced myopia in our patients.

**Table 5.** Visual side effects of hyperbaric oxygen at 2.0 ATA in stroke patients.

| Patient | Glasses | Snellen Chart before HBOT | | Snellen Chart after 40 HBOT Sessions | | Snellen Chart 3 Months after HBOT | |
|---|---|---|---|---|---|---|---|
| | | Left | Right | Left | Right | Left | Right |
| 1 | Yes | 1.08 | 0.96 | 1.08 | 0.96 | 0.96 | 0.84 |
| | No | 0.12 | 0.12 | 0.12 | 0.12 | 0.12 | 0.12 |
| 2 | Yes | 0.84 | 1.2 | 0.9 | 1.0 | 0.84 | 1.2 |
| | No | 0.36 | 0.24 | 0.5 | 0.6 | 0.48 | 0.48 |
| 3 | Yes | - | - | - | - | - | - |
| | No | 0.8 | 0.8 | 0.8 | 0.9 | 0.8 | 0.9 |
| 4 | Yes | - | - | - | - | - | - |
| | No | 0.96 | 0.96 | 0.84 | 0.72 | 0.84 | 0.72 |
| 6 | Yes | 0.36 | 0.48 | 0.72 | 0.72 | 0.7 | 0.7 |
| | No | 0.24 | 0.36 | 0.24 | 0.24 | 0.5 | 0.5 |
| 9 | Yes | 0.96 | 0.72 | 1.08 | 0.72 | 0.96 | 0.72 |
| | No | <0.1 | <0.1 | <0.1 | <0.1 | 0.12 | 0.12 |
| 10 | Yes | 0.6 | 1.08 | 0.6 | 0.84 | 0.6 | 1.08 |
| | No | 0.36 | 0.72 | 0.36 | 0.8 | 0.36 | 0.72 |
| 11 | Yes | - | - | - | - | - | - |
| | No | 0.96 | 1.2 | 0.84 | 1.2 | 0.96 | 1.2 |
| 12 | Yes | - | - | - | - | - | - |
| | No | 0.84 | 0.6 | 0.84 | 1.2 | 1.2 | 1.2 |
| 13 | Yes | - | - | - | - | - | - |
| | No | 0.96 | 1.2 | 0.84 | 0.84 | 0.8 | 1.0 |

### 3.4. Pressure-Related Effects

Patient 10 had difficulties in equalizing and had to pause HBOT; however, the patient resumed HBOT after 7 days and fulfilled the program as scheduled. In this patient, digital otoscopy showed a transitory barotrauma of the eardrum, classified as Teed Grade 2, which normalized after one week.

## 4. Discussion

Our study confirmed that 40 sessions of HBOT are well tolerated in unselected patients with mild-to-moderate residual post-stroke symptoms. Even though potential participants were informed about the experimental value of HBOT in chronic stroke and the requirement for a high level of personal effort and time, our patients were highly motivated to participate in what they considered to be an innovative treatment modality.

Regarding side effects, barotrauma is usually more frequent in elderly patients [8]; however, we did not find an increased risk of barotrauma in our symptomatic post-stroke elderly patients, even after 40 sessions of HBOT. Although HBOT can also cause a myotic shift in elderly patients [9], we did not find a change in visual acuity in any of our patients with chronic stroke following HBOT.

With the exception of two patients whose neurological status remained unchanged (Patients 4 and 6), all other patients experienced an improvement in neurological function following HBOT, as measured with the NIHSS. Interestingly, HBOT had no effect on the RMI. We can only speculate on the reasons for this discrepancy. For instance, it is possible that the RMI is not sensitive enough to detect subtle changes, as most of our patients suffered from mild chronic stroke. Alternatively, the improvements in the NIHSS may have been predominantly impacted by changes to the upper extremities, which would not necessarily have been reflected in the RIM scores.

This was a convenience sample, and we did not select our post-stroke patients, having no advanced imaging information regarding their potential for neuroplasticity. Despite this, we were pleased with the relatively high rate of positive responses to HBOT.

Rosario et al. [10] as well as Schiavo et al. [11] had similar results regarding the feasibility of HBOT at 2.0 ATA in their prospective series on unselected post-stroke patients. They also found that these patients were strongly motivated to collaborate with the physicians in a hyperbaric center dedicated to outpatients. Churchill et al. [12] came to the same conclusion with a protocol involving 60 consecutive HBOT sessions. They noted that even 6 months after HBOT, 90% of their patients would be willing to repeat HBOT.

Several researchers have pointed out the plasticity potential of the post-stroke brain, which is suggestive of the potential benefits of HBOT [13]. They have shown that a penumbra of dormant neurons exists around the stroke area, and these can regain some functionality after HBOT. Indeed, Efrati et al. [4] suggested that the existence of a large penumbra could be an indicator of responsiveness to HBOT. This has been further demonstrated with PET scans [14] and with diverse technology using functional MRI [15].

According to the literature, up to 57% of patients can be expected to show HBOT-induced myopia when using compression rates of 2.2–2.5 ATA. [16]. Using 1.5 ATA, Churchill et al. found only 3 out of 55 patients experienced a myopic shift [12]. In our small study at 2.0 ATA, we did not find any cases of HBOT-induced myopia.

HBOT can cause mild barotrauma in 0.2% [17] to 10% of patients [8]. Our study is in accordance with the literature, with only 1 out of 10 patients experiencing mild barotrauma.

Our prospective study had no control group as the main objective was to determine the feasibility, compliance, and side effects from HBOT in patients with mild-to-moderate residual post-stroke symptoms. Our findings are important for chronic stroke patients who still have neurological deficits that hinder them in daily life, even after they have completed established standard-of-care rehabilitative treatments for an extended period. In their search for further therapeutic means of improving their functional status, HBOT may offer a manageable, safe and effective treatment.

We must also take into account a possible placebo effect, although it is unlikely that a placebo effect would have resulted in such a high degree of clinical improvement or persisting clinical effects. We attribute the improvements experienced by our patient group to an HBOT effect because, while there were some maintenance treatments such as physiotherapy, no patient underwent an adjunctive treatment that could have influenced the results. Eight of the ten patients started HBOT one year or more after their stroke, and it was not expected that their adjuvant therapies would have shown such improvements over the 3-month HBOT period. This is especially true for patient 2, who experienced improvements more than 20 years after their stroke, suggesting that dormant neurons can survive for decades.

### 5. Conclusions

Extensive HBOT treatments in post-stroke patients are feasible in an outpatient setting with minor and reversible side effects, even if patients must commute daily to the hyperbaric facility. HBOT has the potential to improve the clinical status of these patients; however, there is an urgent need to select potential responders with imaging techniques, thereby allowing us to better assess the role of HBOT. This can only be achieved with prospective, controlled studies backed with advanced imaging techniques.

**Author Contributions:** Conceptualization, M.G., J.S. and S.E.; methodology, M.G. and J.S.; validation, M.G., J.S. and S.E.; formal analysis, J.S.; investigation, M.G., J.S., P.S., N.P. and S.E.; resources, M.G., J.S. and S.E.; writing—original draft preparation, J.S.; writing—review and editing, M.G. and S.E.; supervision, M.G.; project administration, M.G.; funding acquisition, J.S. and M.G. All authors have read and agreed to the published version of the manuscript.

**Funding:** Funding was provided by Carbagas, which offered the oxygen necessary for this study. Vifor provided 1000 CHF. Electricity for the hyperbaric chamber, maintenance and liability insurance were all provided by M.G.

**Institutional Review Board Statement:** This study was conducted in accordance with the Declaration of Helsinki and approved by the Ethics Committee Ethikkommission Nordwest und Zentralschweiz, project ID 2017-02131, dated 24 January 2018.

**Informed Consent Statement:** Informed consent was obtained from all subjects involved in the study. Written informed consent was obtained from the patient(s) to publish this paper.

**Data Availability Statement:** Data is contained within the article.

**Acknowledgments:** We thank the included patients for participating in this study. Also our thanks go out to the medical staff of the Hyperbaric Center Basel and Vivianne Hasselmann for her help for study monitoring.

**Conflicts of Interest:** M.G. is the Head of the Hyperbaric Center Basel, and J.S. is the founder of the Hyperbaric Center Basel. Both are involved in daily hyperbaric treatments.

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
