# Peer review of "Hyperbaric Oxygen in Post-Stroke Patients: A Feasibility Study"

_ctn, doi:10.3390/ctn7040041_

Round 1

Reviewer 1 Report

Comments and Suggestions for Authors

Interesting results for this patient group. I hope this article is part of a larger study on this subject and more data can be provided? It's a frequently requested treatment indication in our practice but we have little data to support it usually.

The overall quality of the article can be improved. I've included my notes in a separate file. I'm looking forward to your comments. 

----------------------------------------------------------------

REVIEW

Hyperbaric Oxygen in post STroke patients (HOST): a feasibility study

Jörg Schmutz * , Marco Gelsomino * , Patrick Schmucki * , Nils Peters , Stefan T Engelter

Introduction

- The reasoning to employ HBOT in post-stroke patients (physiology, earlier clinical results, systematic reviews etc.) must be improved; only one reference is provided. Since this is a feasibility study, a clear and persuading argument should be made that HBOT could help in rehabilitation. Otherwise, patients should not be subjected to the therapy, travel etc..

M&M

- Please clarify further if patients definitively stopped rehabilitation, or that they were still undergoing treatment (like weekly physical therapy) during HBOT. This is also a point for the discussion section (influence of other therapies on effect).

- Were tympanic tubes considered when patients failed the pressure test? 0,5 ATA is not a lot of pressure and patients usually develop complaints around 0,8-1,0 ATA.

- Please make sure to clarify that the ATA values given are either “extra” pressure, or include the standard barometric pressure of 1,0 ATA.

- What was the reason to do an ENT interview and otoscopy before and after each session? This is quite intensive for both the patient and the physician, and the clinical significance is probably low.

- Please provide primary and secondary outcome measures for clarity. I.m.o. the compliance and clinical outcomes should be put before side effects in M&M and Results sections.

- What was the procedure with patients who missed sessions or took a break etc.? Please elaborate.

Results

- Please provide standard deviations when presenting means.

- Line 76-79: time from stroke to start HBOT is given twice with different values.

- Neurology paragraph could be improved with some examples of improvement (or, if notable, no improvement)

- In literature the effect of HBOT on visual acuity is mild and always transient. While it is informative, I don’t feel it warrants it’s own table, especially since no changes were detected.

-> Consider creating a paragraph for side effects and put the results of barotrauma, visual acuity and acute oxygen toxicity here.

- Please provide more information on compliance by including missed sessions etc. Did you include side effects like tiredness etc.? In our practice, this is one of the biggest side effects and a major contributor to dropout. 

Discussion

- Overall, the structure and argumentation of this section can be improved.

- Line 118-119: this needs to be analyzed critically. Was there a different measure available that would have detected clinical changes better? If not, why still include mostly mild to moderate severity? Please elaborate (suggestion: make a new paragraph for this point).

- Line 123-125: in such a small sample without a control group, claiming HBOT as the sole reason for improvement is not sound. You state yourself that spontaneous improvement is possible after 1-2 years. What was the expected effect of continued rehabilitation, physical therapy etc. (if applicable)? Possibility of placebo effect? Please elaborate.

- Line 128-133: please elaborate. HBOT can be quite intensive, not even including travel, and the clinical changes are small in your sample. So why is compliance high in this and earlier samples?

- Line 134-140: I feel this section is more appropriate for the previous paragraph (line 121-126), since it provides an argument for improvement post-stroke.

- Please include a paragraph where you critically analyze your own study, and provide counterarguments to possible weaknesses where appropriate. This will strengthen the overall piece.

Table 2

- Column 2; Time to stroke in months: I assume time from stroke to start HBOT? Please provide clear header.

- Change in NIHS score is provided both in Table 2 and Table 3. Consider leaving this out of Table 2 to improve readability.

Table 3

- Consider adding extra column for NIHS score alone

- Patient 2,3,10: “improvement” instead of symbol

- Patient 11: NIHS score given as E?

References

- Reference 1: when citing web pages please provide page title or subject, last update date and date of visit.

- Reference 4: please cite either the web page, or cite authors, title etc. of the referred document (preferably the latter).

- Reference 5: I feel citing scientific literature would be more appropriate to support your statements. If you’re keeping the site, see reference 1 for improvement in citation

Reviewer 2 Report

Comments and Suggestions for Authors

Well described background and methods. Results are clear. Minor improvements in language are suggested. One subject was treated many months after stroke, more than all other subjects. Please comment.

---------------------------------------------------------------------------

1 the main question is a description of responses from 10 patients who completed 40 HBO treatment after discharge from hospital at end of rehab, from stroke. 2 very relevant, as the study follows European standard of HBO treatment and reports acceptable improvements in 8 of 10 patients, after a large variety of time lags from stroke. 3it lists ocular results (no complications) in pts of advance age and support the use of HBO even after several months 4 None. They might stratify patients in early times vs. late times. 5 acceptable conclusions 6 well-chosen references, ready to be accessed by reader 7 there are no figures. The tables are easy to navigate.

Comments on the Quality of English Language

Well described background and methods. Results are clear. Minor improvements in language are suggested. One subject was treated many months after stroke, more than all other subjects. Please comment.

Round 2

Reviewer 1 Report

Comments and Suggestions for Authors

Thank you for the speedy corrections to the article. I feel it reads a lot clearer now. I also agree wtih the reasoning given to not make changes.